# Functional-level Uncertainty Quantification for Calibrated Fine-tuning on LLMs

## Abstract

From common-sense reasoning to domain-specific tasks, parameter-efficient fine tuning (PEFT) methods for large language models (LLMs) have showcased significant performance improvements on downstream tasks. However, fine-tuned LLMs often struggle with overconfidence in uncertain predictions, particularly due to sparse training data. This overconfidence reflects poor epistemic uncertainty calibration, which arises from limitations in the model's ability to generalize with limited data. Existing PEFT uncertainty quantification methods for LLMs focus on the post fine-tuning stage and thus have limited capability in calibrating epistemic uncertainty. To address these limitations, we propose Functional-Level Uncertainty Quantification for Calibrated Fine-Tuning (UQ4CT), which captures and calibrates functional-level epistemic uncertainty during the fine-tuning stage via a mixture-of-expert framework. We show that UQ4CT reduces Expected Calibration Error (ECE) by more than $25\%$ while maintaining high accuracy across 5 benchmarks. Furthermore, UQ4CT maintains superior ECE performance with high accuracy under distribution shift, showcasing improved generalizability.

## 1 Introduction

Large Language Models (LLMs) have revolutionized various domains as general task solvers (Chang et al., 2024). To adapt LLMs for specific downstream tasks or create instruction-following models, fine-tuning have become increasingly important (Houlsby et al., 2019; Hu et al., 2021a; Liu et al., 2022; Ding et al., 2022; 2023). This involves additional training on pre-trained LLMs using a smaller dataset (Zhong et al., 2021; Ren et al., 2022). Through fine-tuning, the model parameters are updated to better adapt to the domain-specific knowledge (Peng et al., 2023). To reduce the computational cost for fine-tuning, Hu et al. (2021a) proposed Low-Rank Adaptation (LoRA), which effectively reduces the parameters required for fine-tuning by introducing low-rank trainable matrices at each layer of the transformer architecture instead of fine-tuning the full model parameters. Li et al. (2024); Wu et al. (2024b) proposed LoRA Mixture-of-Experts (MoE) models which grants better performance while maintaining parameter efficiency.

However, previous studies have shown that fine-tuned LLMs are often overconfident with their predictions (Xiao et al., 2022c; He et al., 2023; Tian et al., 2023; OpenAI, 2023). This resembles poorly calibrated uncertainty (Zhou et al., 2022) due to the sparsity of fine-tuning data. Overconfidence is a crucial problem in safety-related decision making or in fields where data is very limited, such as experimental design, climate science and medical diagnosis (Singhal et al., 2022; Wu et al., 2023a; Lampinen et al., 2023; Li et al., 2022). Thus, methods that enhance uncertainty quantification of fine-tuned LLMs is urgently needed to assure trustworthy predictions for better application.

Established uncertainty quantification methods have been studied in conjunction with the LoRA structure. Monte-Carlo dropout (Gal & Ghahramani, 2016b) interprets dropout in neural networks as approximate Bayesian inference in deep Gaussian processes, allowing uncertainty estimates to be obtained from existing LoRA adapters without modifying them. Checkpoint ensemble (Chen et al., 2017) utilizes predictions from multiple LoRA checkpoints saved during a single fine-tuning process to calibrate uncertainty. Deep ensemble (Lakshminarayanan et al., 2017; Wang et al., 2023; Zhai et al., 2023a) combines the predictions from multiple LoRA adapters for better uncertainty calibration. Laplace-LoRA (Yang et al., 2024a) applies Bayesian inference via Laplace approximation to the LoRA parameters after fine-tuning, resulting in improved calibration and uncertainty estimates.

Although these methods have demonstrated improved uncertainty estimations, they all utilize a fixed set of LoRA parameters fine-tuned over the entire downstream task dataset. The point estimates of parameters have very limited capabilities in capturing epistemic uncertainty, while direct calibration of epistemic uncertainty over the entire LoRA parameter space is an ideal but not practical approach.

Therefore, we propose Functional-Level Uncertainty Quantification for Calibrated Fine-Tuning (UQ4CT) to calibrate the functional-level epistemic uncertainty via the LoRA MoE architecture. We propose a functional perspective on LoRA parameters where we treat the LoRA experts as basis functions and consider the more complex, prompt dependent functions as mixtures of those basis functions. On top of learning the parameters in the LoRA experts, UQ4CT also trains a prompt-dependent LoRA mixture to form a calibrated distribution over the functional space. The LoRA experts capture different functional relationships in the fine-tuning data throughout training, and the MoE routers dynamically select these functional bases conditioned on the input. The selection process models the functional level epistemic uncertainty, and consequently captures the uncertainty in the output space. We calibrate functional level epistemic uncertainty to align with predictive correctness during training time. This significantly improves uncertainty estimations of the model on its predictions without compromising the accuracy. To summarize, our contributions include:

- A novel epistemic uncertainty quantification approach with Mixture-of-Experts architecture during fine-tuning stage to model functional level epistemic uncertainty and align with predictive correctness, which mitigates overconfidence issue and improves generalizability.

- A novel training calibration loss function incorporating predictive correctness to calibrate the prompt-dependent LoRA mixture for better uncertainty estimation.

- More than $25\%$ Expected Calibration Error reduction on $4$ common-sense reasoning tasks and $1$ domain-specific question answering task, superior ECE performance under distribution shift scenarios on $2$ common-sense reasoning tasks and $4$ domain-specific question answering tasks without compromising accuracy.

## 2 PRELIMINARIES

### 2.1 LOW-RANK ADAPTATION (LORA)

LLMs have numerous large weight matrices to perform matrix multiplication, denoted as $\mathbf{W}_0 \in \mathbb{R}^{n_{\text{out}} \times n_{\text{in}}}$ that maps inputs $\mathbf{x}$ to outputs $\mathbf{h}$. Hu et al. (2021a) proposes LoRA, which fixes $\mathbf{W}_0$ and introduces a low-rank perturbation $\Delta\mathbf{W}$ to the weight matrix:

$$\mathbf{h} = \mathbf{W}_0\mathbf{x} + \Delta\mathbf{W}\mathbf{x} = \mathbf{W}_0\mathbf{x} + \mathbf{BA}\mathbf{x}. \tag{1}$$

Here, $\Delta\mathbf{W}$ is calculated as the product of two matrices, $\mathbf{B} \in \mathbb{R}^{n_{\text{out}} \times n_{\text{lr}}}$ and $\mathbf{A} \in \mathbb{R}^{n_{\text{lr}} \times n_{\text{in}}}$ where $n_{\text{lr}}$ is significantly smaller than $n_{\text{in}}$ or $n_{\text{out}}$. For example, we use $n_{\text{lr}} = 32$ while $n_{\text{in}} = n_{\text{out}} = 4096$ for the Llama2-7b model (Touvron et al., 2023c). Therefore, the total number of LoRA parameters for this $\Delta\mathbf{W}$ is $n_{\text{lr}}(n_{\text{in}} + n_{\text{out}})$, which is far smaller than the parameter count of the full matrix, $n_{\text{in}}n_{\text{out}}$. One of the key motivations of incorporating LoRA to fine-tune LLMs is the vast amount of memory cost reduction compared with fine-tuning on the full model. For an LLM with 7 billion parameters, maintaining the average gradient and average squared gradients for optimization multiplies the memory required by a factor of 3 compared to simply loading model weights. LoRA greatly mitigates this memory cost as the tripled memory consumption only applies to LoRA adapters.

### 2.2 MIXTURE OF EXPERTS (MOE)

LoRA Mixture-of-Experts (Li et al., 2024; Wu et al., 2024b) is a efficient approach to scale the number of parameters while maintaining the same computational bounds. LoRA MoE utilizes the top-k router to assign each token to the LoRA experts (Lepikhin et al., 2020). The router is a linear layer that maps the input hidden state $\mathbf{h}$ to a probability distribution of candidate experts.

Let $\mathbf{h}_i^\ell \in \mathbb{R}^{1 \times d}$ $(1 \leq i \leq s, 1 \leq \ell \leq L)$ denote the output hidden state of the $i$-th token at the $\ell$-th layer of the LLM, where $L$ is the number of LLM layers and $d$ is the hidden dimension. With $\mathbf{W}_r^\ell$ as

the trainable router weight at layer $\ell$, the top-k gate router chooses $k$ experts with highest probability given a hidden state $\mathbf{h}_i^\ell$:

$$R^\ell(\mathbf{h}_i^\ell) = \text{KeepTop-k}(\text{Softmax}(\mathbf{W}_r^\ell \cdot \mathbf{h}_i^\ell)). \qquad (2)$$

Finally, we obtain the final MixLoRA prediction with:

$$\text{MixLoRA}(\mathbf{h}^\ell) = \sum_{k=1}^{K} R^\ell(\mathbf{h}^\ell)_k E_k^\ell(\mathbf{h}^\ell), \quad E_k^\ell(\mathbf{h}^\ell) = \mathbf{W}^\ell \cdot \mathbf{h}^\ell + \mathbf{B}_i^\ell \mathbf{A}_i^\ell \cdot \mathbf{h}^\ell \qquad (3)$$

where $\mathbf{W}$ is the pretrained weights of the feed-forward network (FFN) layer and $\mathbf{B}_i^\ell \mathbf{A}_i^\ell$ is the i-th LoRA expert.

## 2.3 ALEATORIC AND EPISTEMIC UNCERTAINTIES

In machine learning models, uncertainty can be categorized into aleatoric (data-wise) and epistemic (model-wise) uncertainty (Hora, 1996; Hüllermeier & Waegeman, 2021). For LLMs, aleatoric uncertainty arises from the inherently ambiguous and context dependent nature of natural languages where a single phrase or sentence can have multiple valid interpretations in different contexts. Epistemic uncertainty is introduced by the model's lack of knowledge due to limited learning capabilities, suboptimal modeling or sparse training data.

Epistemic uncertainty is highly related to several well-known limitations of generative models. For example, it has been observed that when an LLM is pretrained on a diverse range of text data, it is generally well-calibrated, i.e. the predicted probability of the next token generally aligns with what is observed in real text. However, after fine-tuning or alignment with human preferences, the calibration error deteriorates (Zhao et al., 2021; Achiam et al., 2023a). A related phenomenon is forgetting, where the performance of a fine-tuned LLM diminishes on tasks outside the scope of the target downstream task (Lin et al., 2023; Luo et al., 2023).

Motivated by these observations, we explore functional-level epistemic uncertainty in generative models and aim to develop metrics that assess model performance on specific problem instances to fine-tune the parameter mixture.

## 3 METHODOLOGY

The high level goal of UQ4CT is to balance the exploration and exploitation of different LoRA experts during fine-tuning. In particular, we incorporate the functional-level epistemic uncertainty (FEU) to calibrate the prompt-dependent parameter mixture with LoRA MoE.

Assume that our answer $a$ is generated via a mixture of mechanisms or models $M$, conditioning on the input prompt $x$. Assume that $e(a)$ is an embedding of $a$ so that least squares distance is a natural distance on the space of $e(a)$. For an expressive enough model class $\mathcal{M}$, and a calibrated distribution $P(M|x)$ over the model class, we can measure the deviation of the generated answer to the "ideal" one $a_* = f_*(x)$ as:

$$\mathbb{E}_{M \sim P(M|x)} \left[ \mathbb{E}_{a \sim P(a|M,x)} \left[ \|e(a) - e(a_*)\|^2 \right] \right]$$
$$= \mathbb{E}_{P(M|x)} \underbrace{\left[ \mathbb{E}_{P(a|M,x)} \left[ \|e(a) - \mathbb{E}_{P(a|M,x)}[e(a)]\|^2 \right] \right]}_{\text{Aleatoric Uncertainty}} + \underbrace{\mathbb{E}_{P(M|x)} \left[ \|\mathbb{E}_{P(a|M,x)}[e(a)] - e(a_*)\|^2 \right]}_{\text{Epistemic Uncertainty}}.$$

$$(4)$$

Note that we hereby quantify uncertainty as a function of the input prompt $x$, since the distribution of model $M$ conditions on $x$. We hence name the task "functional-level uncertainty quantification".

## 3.1 FUNCTIONAL-LEVEL EPISTEMIC UNCERTAINTY

Motivated by the decomposition in Eq. (4) for least squares loss, we may consider a general distance $d$, and define epistemic uncertainty that characterizes the variation caused by model training procedure. Specifically, we focus on the variation introduced in the model fine-tuning stage of LLMs.

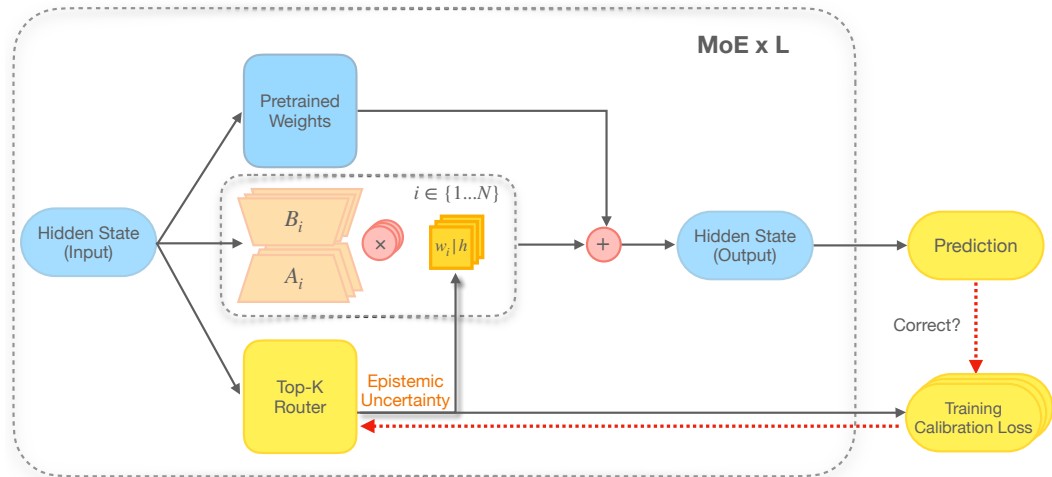

Figure 1: Mixture of Experts (MoE) architecture to capture and calibrate functional-level epistemic uncertainty. Experts $B_{1...N}A_{1...N}$ capture functional relationships in the data throughout fine-tuning, the weights $w_{1...N}|h$ quantify the uncertainty in selecting these functional bases conditioned on the input hidden state $h$, which is the semantic representation of the input token $x$. In the UQ4CT workflow, we align this uncertainty with predictive correctness. When the router makes a correct prediction, the loss reinforces this decision, thereby increasing the router confidence in its selection, which aligns with a lower epistemic uncertainty. Conversely, when the router makes an incorrect prediction, the loss penalizes this selection, potentially causing the router to distribute its probabilities more broadly across experts, which is indicative of higher epistemic uncertainty.

Mathematically, given prompt $x$, we consider the following definition of epistemic uncertainty:

$$\text{Epistemic Uncertainty} = \mathbb{E}_{M \sim P(M|x)}\mathbb{E}_{M' \sim P(M'|x)}\left[\|\mathbb{E}_{P(a|M,x)}[e(a)] - \mathbb{E}_{P(a'|M',x)}[e(a')]\|^2)\right]. \tag{5}$$

Here, $a'$ represents the ground truth output sampled from the ideal MoE model $M'$ conditioned on the prompt $x$. The epistemic uncertainty measures the least squares distance between $e(a)$ and $e(a')$ from current mixture distribution $P(M|x)$ and the ideal mixture distribution $P(M'|x)$.

### 3.2 QUANTIFYING FEU WITH LoRA MoE FRAMEWORK

We quantify the functional-level epistemic uncertainty (FEU) with the MoE architecture, which is represented by the embedding $e(a)$ in Eq. (5). As shown in Figure 1, the LoRA experts $B_{1...N}A_{1...N}$ capture important functional relationships in the data during fine-tuning. We treat these functions represented by the LoRA experts as basis functions $f_{1...N}$ and define them as follows:

$$f_1, f_2, \ldots, f_N = \{B_1A_1, B_2A_2, \ldots, B_NA_N\}. \tag{6}$$

Conditional on the input prompts, the more complex functional relationships that recursively map inputs to outputs are represented as linear combinations (mixture of experts) of the basis functions.Uncertainty quantification in this functional space reduces to quantifying the uncertainty of the weights over the basis functions. In particular, the weights $w_{1...N}|h$ from the top-k router dynamically selects these basis functions conditioned on the input hidden state $h$:

$$h = \sum_{i=1}^{N}(w_i|h) \cdot f_i = \sum_{i=1}^{N}(w_i|h) \cdot (B_iA_i). \tag{7}$$

The top-k weights that produce the final output hidden state quantify the functional level epistemic uncertainty in the function selection.

In the mixture of LoRA experts architecture, we follow the routing mechanisms of the MoE layers as in Eq. (2) and (3). Specifically, we employ top-2 gate routers, which chooses the 2 experts with

highest probability given a hidden state $\mathbf{h}_i^\ell$:

$$R^\ell(\mathbf{h}_i^\ell) = \text{KeepTop-2}(\text{Softmax}(\mathbf{W}_r^\ell \cdot \mathbf{h}_i^\ell)). \tag{8}$$

Given an input prompt $x$ with length $s$, we model functional-level epistemic uncertainty (FEU) by aggregating $R^\ell(\mathbf{h}_i^\ell)$ over both layer dimension and sequence dimension:

$$\text{FEU}(x) = \frac{1}{s}\sum_{i=1}^{s}\left[\frac{1}{L}\sum_{\ell=1}^{L}R^\ell(\mathbf{h}_i^\ell)\right] \tag{9}$$

### 3.3 Training Calibration Loss

We then calibrate the FEU model of the epistemic uncertainty against predictive accuracy. Specifically for the MoE top-k routers, we design the following calibration loss for training:

$$\mathcal{L}_{\text{cal}} = \|\mathbb{1}\{\text{MixLoRA}(x) = y^*\} - \text{FEU}(x)\|^2, \tag{10}$$

where the first term is an indicator function of whether the model prediction matches the ground truth $y^*$ given the prompt $x$. Here, the indicator function resembles $\mathbb{E}_{P(a'|M',x)}[e(a')]$ in Equation 5, where the ground truth $y^*$ is $a'$ and the indicator function maps the predictive correctness to a confidence space $e \in [0, 1]$. We employ a one-hot definition of the ground truth confidence. When the prediction from current mixture model matches the ground truth, the ground truth confidence is 1. Otherwise, when the predictions do not match, the ground truth confidence is 0.

As shown in Figure 1, this term effectively promotes expert exploitation for correct predictions and expert exploration for incorrect predictions by directly calibrating the functional level epistemic uncertainty to align with the predictive correctness. Ideally, when the $N$ LoRA experts together capture all the functional relationships across the data distribution with cross entropy during fine-tuning, our proposed loss $\mathcal{L}_{cal}$ also finds proper mixture of LoRA experts conditioned on the input $x$ by conditionally promoting expert exploitation and exploration. This allows the model to select correct functional relationships regarding $x$ to generate an output that better matches the data distribution, which grants calibrated uncertainty estimations.

Load balancing is a common technique to ensure even exploitation across experts with the MoE architecture (Fedus et al., 2022). We follow the load balancing loss $\mathcal{L}_b$ proposed by (Li et al., 2024) and define our loss function as:

$$\mathcal{L} = \mathbf{CE} + \alpha \cdot \mathcal{L}_b + \beta \cdot \mathcal{L}_{\text{cal}}, \tag{11}$$

where $\mathbf{CE}$ represents cross entropy loss, $\alpha$ and $\beta$ are the hyperparameters of two auxiliary terms. Details about $\mathcal{L}_b$ can be found in Appendix A.1.

## 4 Related Work

### 4.1 Parameter-Efficient Fine Tuning for LLMs

Large Language Models (Brown et al., 2020; Chowdhery et al., 2022; Hoffmann et al., 2022; Touvron et al., 2023a;d) have shown impressive abilities in handling various natural language processing tasks. Building on these advances, instruction fine-tuning(Chung et al., 2022; Iyer et al., 2022; Zheng et al., 2024) has enhanced LLMs' capacity to comprehend and follow human instructions, forming the core of modern conversational AI systems(Wu et al., 2023b; Achiam et al., 2023b). However, as LLMs increase in size, the fine-tuning process demands much more time and memory.

To address these challenges, several strategies have been proposed, including parameter-efficient fine-tuning (PEFT)(Mangrulkar et al., 2022), model distillation(Liu et al., 2023; Xiao et al., 2023), quantization(Frantar et al., 2022; Xiao et al., 2022a), and pruning(Frantar & Alistarh, 2023; Ma et al., 2023). Among these, LoRA(Hu et al., 2021b), which leverages low-rank matrix decomposition of linear layer weights, is a widely adopted PEFT technique that boosts model performance without adding computational costs during inference. For example, VeRA(Kopiczko et al., 2023) introduces learnable scaling vectors to modify shared pairs of frozen random matrices across layers, while FedPara(Hyeon-Woo et al., 2021) focuses on low-rank Hadamard products for federated learning

settings. Tied-Lora(Renduchintala et al., 2023) applies weight tying to further minimize the number of trainable parameters. AdaLoRA(Zhang et al., 2023) uses Singular Value Decomposition (SVD) to prune less important singular values for efficient updates, and DoRA(Liu et al., 2024) separates pre-trained weights into magnitude and direction components, applying LoRA to update the directional component during fine-tuning, thus reducing the number of parameters to be trained.

## 4.2 MIXTURE OF EXPERTS

The Mixture-of-Experts concept(Jacobs et al., 1991), introduced as early as 1991, presented a novel supervised learning framework where multiple networks (experts) specialize in handling distinct subsets of training data. Modern MoE variants adapt this by modifying the traditional feed-forward sub-layer within transformer blocks, incorporating sparsely activated LoRA experts, which allows for significant expansion in model width without a proportional increase in computational overhead.

Different MoE architectures have since emerged, distinguished by their expert sampling and routing strategies. For example, LLaVA-MoLE(Chen et al., 2024) improves token routing to domain-specific experts within transformer layers, reducing data conflicts and consistently outperforming standard LoRA baselines. Other MoE-based approaches include MoRAL(Yang et al., 2024b), which focuses on efficiently adapting LLMs to new domains and tasks for lifelong learning, and LoRAMoE(Dou et al., 2024), which incorporates LoRAs via a router network to mitigate the issue of world knowledge forgetting. PESC(Wu et al., 2024a) transforms dense models into sparse ones through an MoE structure, lowering computational and GPU memory requirements. MoE-LoRA(Luo et al., 2024) introduces a new parameter-efficient MoE method using Layer-wise Expert Allocation (MoLA) for transformer models, while MoCLE(Gou et al., 2023) activates task-specific model parameters based on instruction clusters. MixLoRA (Li et al., 2024) implements a high-throughput framework for LoRA MoE training and inference process, constructing LoRAs as stochastic experts to reduce computational overhead while expanding model capacity.

Despite the performance improvements these architecture advancements have brought, the overconfidence problem of fine-tuned models is lacking attention (Xiao et al., 2022c; He et al., 2023; Tian et al., 2023; OpenAI, 2023). Enhancing the uncertainty estimation capabilities of these models is fundamental toward more reliable, interpretable and trustworthy applications of LLMs.

## 4.3 UNCERTAINTY QUANTIFICATION IN LLMS

Uncertainty quantification has garnered substantial attention in various tasks and domains within neural networks(Gal & Ghahramani, 2015; Gal & Ghahramani, 2016a; Malinin & Gales, 2018; Ovadia et al., 2019; Malinin et al., 2021; Lin et al., 2022; Kuhn et al., 2023; Lin et al., 2023). This focus extends to LLMs, where the precise quantification of prediction uncertainty has become a critical area of research(Xiao et al., 2022b; Lin et al., 2022; Mielke et al., 2022; Chen & Mueller, 2023; Duan et al., 2023; Huang et al., 2023). LLMs, particularly in generative tasks, pose unique challenges, especially when it comes to measuring the uncertainty of their outputs(Liu et al., 2019; Malinin & Gales, 2021; Kuhn et al., 2023; Lin et al., 2023). The distinction between aleatoric and epistemic uncertainty was recently examined in the context of LLMs(Hou et al., 2023), though this was approached by ensembling model inputs rather than model instances and did not address fine-tuning tasks specifically.

Existing works have investigated the application of ensembling in fine-tuning LLMs for uncertainty quantification. Gleave & Irving (2022); Sun et al. (2022) focus on uncertainty estimation in full model fine-tuning, while this approach inherently incurs significant memory overhead. Wang et al. (2023); Zhai et al. (2023b); Balabanov & Linander (2024) explore the use of LoRA ensembles for uncertainty estimation in LLMs. Yang et al. (2024a) applies a post-hoc Laplace approximation Mackay (1992) to model LoRA parameters after fine-tuning. BatchEnsemble(Wen et al., 2020), introduces component-specific rank-1 matrices as multiplicative modifications to a base model. Though this method has been applied to LLMs, it has been used in the pre-training phase rather than fine-tuning(Tran et al., 2022). None of these methods provide calibrations on epistemic uncertainty, which is crucial to mitigate overconfidence in the fine-tuning stage given the sparse dataset.

## 5 EXPERIMENTS

### 5.1 DATASETS

We include 5 multiple-choice question answering benchmarks to evaluate UQ4CT: OpenBookQA (OBQA, Mihaylov et al. (2018)), ARC-Easy (ARC-E) and ARC-Challenge (ARC-C) from AI2 Reasoning Challenge (Clark et al., 2018), BOOLQ (Clark et al., 2019) and ClimateQA, an expert-annotated domain specific benchmark for climate science. We also use computer science, law, medication and engineering subsets from MMLU dataset(Hendrycks et al., 2020) to evaluate performance under distribution shift. We fine-tune on the publicly available training split and test on the validation split from these benchmarks to evaluate model performance. We report the average model performance over 3 random runs and the standard deviations in the subscript.

### 5.2 EXPERIMENT SETUP

We implement UQ4CT with PyTorch (Paszke et al., 2019), extending the MixLoRA repository in (Li et al., 2024) and compare the average performance in three random runs and report the mean and standard deviation with following baselines. We use the publicly available LLaMA-2-7B-hf (Touvron et al., 2023c) as our base model. In particular, we apply MixLoRA to query, key, value and output layers, together with the feed-fowrward networks in LLaMA-2-7B-hf (gate layer, down layer and up layer). Details are provided in Appendix A.4

- **LoRA**(Hu et al., 2021a). We use standard LoRA fine-tuning as lower performance bound.
- **Monte Carlo (MC) Dropout**(Gal & Ghahramani, 2016b) keeps dropout on at both training and testing time. By performing multiple forward passes, MC dropout randomly shuts down a portion of model nodes, producing ensemble-alike predictions. To combine LoRA fine-tuning with MC dropout, we apply dropout on the input of the LoRA adapter, following the implementation of Mangrulkar et al. (2022).
- **Deep Ensemble**(Lakshminarayanan et al., 2017) averages the predictions from each ensemble member which have been trained with varying random initialization. We combine deep ensemble with LoRA by fine-tuning 3 randomly initialized LoRAs together and ensembling their output as final predictions.
- **Laplace-LoRA (LA)**(Yang et al., 2024a) applies a post-hoc Laplace approximation on fine-tuned LoRA parameters for better uncertainty estimation.
- **MixLoRA**(Li et al., 2024) incorporates LoRAs via a router network to reduce computational overhead while expanding model capacity. We add this as a baseline to resemble plain LoRA MoE model performance.

**Evaluation Metrics.** We measure the prediction accuracy on the validation set for all 5 tasks. For uncertainty calibration, we incorporate expected calibration error (ECE, Guo et al. (2017)) with 15 bins, which measures the alignment between predicted probabilities and empirical accuracy. We also investigate model performance under distribution shift to ensure the model has predictable behavior when given data from other domains as this is a crucial component for real-world applications. Specifically, we test models fine-tuned on OBQA dataset with 4 domain-specific MMLU subtask ensembles focusing on different professionalities and ARC-E/C datasets to approximate larger and smaller distribution shifts. Metric details are provided in Appendix A.5.

### 5.3 RESULTS

We assess the prediction accuracy and uncertainty calibration of models under both in-distribution and distribution shift scenarios. The in-distribution scenario examines the alignment of the fine-tuned model on the target downstream task, while the distribution shift scenario evaluates the generalizability of the model on novel tasks beyond the fine-tuned domain. These two scenarios combined provides a comprehensive assessment of model robustness in real-world applications where it is essential for the model to excel on its primary task while maintaining the ability to effectively handle unforeseen or out-of-distribution inputs.

Table 1: Performance comparison of different methods fine-tuned with LlaMA2-7B across $4$ common sense reasoning tasks and a domain-specific task. UQ4CT shows substantial ECE improvements while maintaining high accuracy.

| Metrics | Methods | BoolQ | ARC-E | ARC-C | OBQA | ClimateQA |
|---|---|---|---|---|---|---|
| ACC ↑ | LoRA | $69.5_{1.93}$ | $74.8_{1.39}$ | $53.8_{0.6}$ | $72.1_{0.87}$ | $59.9_{2.13}$ |
| | MC Drop | $66.8_{3.66}$ | $76.8_{1.30}$ | $50.9_{2.01}$ | $74.8_{1.34}$ | $58.2_{2.11}$ |
| | Ensemble | $66.2_{3.7}$ | $71.2_{1.0}$ | $47.5_{0.57}$ | $75.5_{1.4}$ | $59.6_{6.9}$ |
| | LA | $68.7_{1.32}$ | $74.6_{2.11}$ | $51.4_{0.83}$ | $70.8_{1.24}$ | $55.2_{3.29}$ |
| | MixLoRA | $71.5_{1.05}$ | $77.7_{2.27}$ | $54.3_{1.07}$ | $75.5_{2.91}$ | $61.6_{1.76}$ |
| | UQ4CT | $73.5_{0.52}$ | $76.6_{1.30}$ | $52.8_{1.77}$ | $77.3_{1.36}$ | $63.3_{1.74}$ |
| ECE ↓ | LoRA | $11.9_{0.78}$ | $11.9_{2.04}$ | $19.4_{4.75}$ | $10.2_{1.07}$ | $14.3_{0.64}$ |
| | MC Drop | $12.2_{0.85}$ | $11.9_{1.99}$ | $19.8_{4.85}$ | $10.9_{0.24}$ | $14.3_{0.56}$ |
| | Ensemble | $7.28_{2.3}$ | $9.1_{1.49}$ | $10.23_{1.39}$ | $8.83_{2.35}$ | $13.5_{3.29}$ |
| | LA | $17.1_{1.72}$ | $16.6_{3.7}$ | $18.1_{0.5}$ | $17.2_{1.2}$ | $12.6_{1.9}$ |
| | MixLoRA | $7.88_{2.09}$ | $9.09_{0.81}$ | $10.74_{1.07}$ | $12.9_{1.99}$ | $12.5_{1.32}$ |
| | UQ4CT | $\mathbf{2.3_{0.82}}$ | $\mathbf{6.0_{0.2}}$ | $\mathbf{6.1_{1.11}}$ | $\mathbf{5.0_{1.15}}$ | $\mathbf{8.1_{0.52}}$ |

### 5.3.1 IN-DISTRIBUTION PERFORMANCE

We first evaluate UQ4CT and baseline models fine-tuned on the $4$ common sense reasoning tasks and the climate question answering task under the in-distribution scenario, where models are trained and evaluated on different splits of the same dataset. Note that one of the key advantages of UQ4CT is that uncertainty calibration happens during the fine-tuning stage with little computational overhead, while other UQ methods require repetitive sampling or other post-hoc complexities.

As shown in Table 1, UQ4CT demonstrates notable improvements in uncertainty calibration across a variety of tasks. Across all benchmarks, UQ4CT maintains competitive accuracy (ACC) compared to the baseline methods. For example, on the BoolQ and ClimateQA datasets, UQ4CT achieves accuracy rates of 73.5% and 63.3%, respectively. This empirically demonstrates that UQ4CT is capable of maintaining high accuracy with uncertainty calibration, which assures the gain in UQ performance does not compromise accuracy.

The most substantial performance improvement is observed in the reduction of Expected Calibration Error (ECE). UQ4CT consistently outperforms other methods, reducing ECE by more than $25\%$ on average across the evaluated benchmarks. Unlike other methods where the ECE performance is worsened on the more challenging ARC-C benchmark, UQ4CT achieves an ECE score of 6.1, showcasing the effectiveness of the calibration.

In addition to experiments on LLaMA-2-7B in the main text, we also present additional experiments on fine-tuning Mistral-7B in Appendix A.2 for more comprehensive evaluation of our method. For both LLaMA-2-7B and Mistral-7B models, UQ4CT consistently shows substantial improvements in uncertainty calibration across various tasks. The improvements are critical in applications where the model's confidence must align with its predictive accuracy given limited data, particularly in safety-critical and domain-specific tasks.

### 5.3.2 PERFORMANCE UNDER DISTRIBUTION SHIFT

Due to the sparse nature of the fine-tuning data, real world deployment of LLMs often requires the model to be robust to out-of-distribution knowledge (Ouyang et al., 2022; Touvron et al., 2023b;c). Therefore, we evaluate the performance of UQ4CT along with other baseline models fine-tuned on the OBQA dataset under smaller and larger distribution shift scenarios. Similar to the dataset setup in (Yang et al., 2024a), we use ARC-C and ARC-E dataset to simulate smaller distribution shift because the ARC dataset has similar domain focus on general science reasoning, but is generally more challenging and covers a broader range of scientific topics than OBQA. For larger distribution shift, we ensemble the domain-specific MMLU subtasks into $4$ benchmarks focusing on different professionalities: Computer Science (CS), Engineering (Eng), Law and Health. These tasks have very broad coverage of the domain task at various knowledge levels ranging from elementary school

Table 2: Performance comparison of different methods fine-tuned on OBQA dataset with LlaMA2-7B across 2 smaller distribution shift (DS) tasks and 4 larger distribution shift tasks. UQ4CT shows substantial ECE improvements while maintaining high accuracy.

| Metrics | Methods | ID | Smaller DS | | Larger DS | | | |
|---------|---------|------|------------|------|-----|-----|-----|--------|
| | | OBQA | ARC-C | ARC-E | CS | Eng | Law | Health |
| ACC ↑ | LoRA | $72.1_{0.87}$ | $58.6_{1.93}$ | $66.5_{3.38}$ | $35.5_{2.35}$ | $30.8_{1.72}$ | $34.9_{1.41}$ | $39.1_{1.52}$ |
| | MC Drop | $74.8_{1.34}$ | $58.7_{2.07}$ | $66.6_{3.30}$ | $36.0_{1.69}$ | $30.3_{2.25}$ | $35.1_{0.86}$ | $39.1_{1.35}$ |
| | Ensemble | $75.5_{1.4}$ | $57.7_{0.78}$ | $69.1_{0.48}$ | $36.7_{2.18}$ | $30.3_{1.13}$ | $35.3_{1.02}$ | $39.9_{1.89}$ |
| | LA | $70.8_{1.24}$ | $58.7_{0.58}$ | $67.9_{0.41}$ | $33.7_{1.22}$ | $29.6_{1.32}$ | $35.4_{0.75}$ | $38.5_{1.61}$ |
| | MixLoRA | $75.5_{2.91}$ | $58.5_{1.44}$ | $69.2_{1.02}$ | $35.2_{2.92}$ | $30.3_{0.98}$ | $35.9_{0.43}$ | $40.6_{1.13}$ |
| | UQ4CT | $77.3_{1.36}$ | $58.8_{1.06}$ | $65.8_{1.31}$ | $36.2_{1.24}$ | $34.1_{2.31}$ | $35.8_{1.01}$ | $40.0_{1.24}$ |
| ECE ↓ | LoRA | $10.2_{1.07}$ | $16.7_{2.28}$ | $13.3_{2.48}$ | $29.7_{2.69}$ | $32.3_{1.85}$ | $29.2_{3.08}$ | $31.0_{2.13}$ |
| | MC Drop | $10.9_{0.24}$ | $16.7_{2.20}$ | $13.2_{2.21}$ | $23.2_{2.32}$ | $31.6_{1.64}$ | $28.0_{2.93}$ | $25.9_{2.27}$ |
| | Ensemble | $8.83_{2.35}$ | $15.1_{1.09}$ | $11.1_{0.99}$ | $22.4_{1.32}$ | $28.5_{2.13}$ | $29.0_{1.37}$ | $24.5_{0.39}$ |
| | LA | $17.2_{1.2}$ | $16.2_{0.5}$ | $24.4_{0.42}$ | $28.6_{2.61}$ | $30.5_{1.43}$ | $29.5_{1.83}$ | $30.7_{1.2}$ |
| | MixLoRA | $12.9_{1.99}$ | $19.0_{1.88}$ | $14.5_{2.57}$ | $26.4_{3.25}$ | $33.7_{1.87}$ | $30.3_{2.27}$ | $28.3_{1.07}$ |
| | UQ4CT | $\mathbf{5.0_{1.15}}$ | $\mathbf{8.9_{3.46}}$ | $\mathbf{6.5_{1.85}}$ | $\mathbf{19.6_{2.90}}$ | $\mathbf{23.1_{1.17}}$ | $\mathbf{25.9_{3.43}}$ | $\mathbf{21.9_{3.49}}$ |

to professionals. This domain-specificity demonstrates larger distribution shift from OBQA, which is a general common sense reasoning task. Details of the ensemble is provided in Appendix. A.6.

The distribution shift evaluations are provided in Table 2. UQ4CT provides substantial improvements in terms of ECE while maintains similar accuracy for both smaller and larger distribution shift scenarios. For smaller distribution shifts, UQ4CT shows comparable ECE performance as the in-distribution scenario. For the more challenging larger distribution shifts, UQ4CT still achieves the best ECE performance among all baseline models. Note that UQ4CT also achieves competitive prediction accuracy across all domain-specific tasks. This empirically shows that our proposed alignment of the functional epistemic uncertainty with predictive correctness improves generalizability and mitigating the overconfidence problem on the fine-tuned model.

## 5.4 ABLATION STUDY

In this section, we conduct ablation studies to investigate the effectiveness of our designed calibration loss, $\mathcal{L}_{cal}$. We first evaluate the incremental weighting performance of the calibration term, which investigates the effectiveness of $\mathcal{L}_{cal}$ at the early stage of fine-tuning. Then, we perform a sensitivity test, where we explore the overall performance impact of $\mathcal{L}_{cal}$. We also conduct an ablation study on the impact of active LoRA experts in Appendix A.3.

### 5.4.1 INCREMENTAL WEIGHTING ON CALIBRATION TERM

Due to the random initialization of LoRA experts, the predictions during early fine-tuning stage are likely to be incorrect as the model has little knowledge on the functional relationships regarding the data. Thus, it is intuitive to incrementally increase the weight parameter $\beta$ over the calibration term $\mathcal{L}_{\text{cal}}$ in the training loss for the LoRA experts to learn before calibration. We conduct this study by incrementally increase $\beta$ from 0 to 1 within 50 gradient steps during the early stage of fine-tuning:

$$\beta = \min\left\{1, \frac{\text{current\_grad\_step}}{50}\right\}. \tag{12}$$

We choose 50 gradient steps from our observation that training loss generally stabilizes after 50 gradient steps, indicating the LoRA experts have learned some functional relationships from data.

As shown in Table 3, the incremental loss has significantly worse ECE performance across all tasks. This demonstrates the advantage of uncertainty calibration even in the early stage. In the beginning, the lack of functional relationships on the training data in LoRA experts lead to high epistemic uncertainty. Thus, UQ4CT encourages exploration over all LoRA experts while UQ4CT_Incremental lacks it due to the small weighting in the beginning.

Table 3: Performance comparison of UQ4CT with and without incremental weighting. Incremental weighting has worse ECE performance while maintains similar accuracy.

| Metrics | Methods | BoolQ | ARC-E | ARC-C | OBQA | ClimateQA |
|---------|---------|-------|-------|-------|------|-----------|
| ACC $\uparrow$ | UQ4CT | $73.5_{0.52}$ | $76.6_{1.30}$ | $52.8_{1.77}$ | $77.3_{1.36}$ | $63.3_{1.74}$ |
| | UQ4CT_Incremental | $72.0_{0.19}$ | $75.4_{0.81}$ | $54.6_{0.95}$ | $77.6_{0.43}$ | $60.2_{3.17}$ |
| ECE $\downarrow$ | UQ4CT | $\mathbf{2.3_{0.82}}$ | $\mathbf{6.0_{0.2}}$ | $\mathbf{6.1_{1.11}}$ | $\mathbf{5.0_{1.15}}$ | $\mathbf{8.1_{0.52}}$ |
| | UQ4CT_Incremental | $4.0_{0.16}$ | $9.8_{1.51}$ | $13.8_{2.08}$ | $10.3_{1.73}$ | $12.2_{0.88}$ |

Table 4: Performance of UQ4CT with varying $\beta$ value on OBQA dataset. Prediction accuracy and uncertianty alignment increases with $\beta$, highlighting the effectiveness of the calibration term.

| $\beta$ | ACC $\uparrow$ | ECE $\downarrow$ |
|---------|----------------|------------------|
| 0 | $75.5_{2.91}$ | $12.9_{1.99}$ |
| 0.2 | $76.0_{0.6}$ | $7.47_{0.78}$ |
| 0.5 | $75.9_{0.31}$ | $7.82_{0.93}$ |
| 0.8 | $76.6_{0.4}$ | $5.94_{1.16}$ |
| 1 | $\mathbf{77.3_{1.36}}$ | $\mathbf{5.0_{1.15}}$ |

### 5.4.2 SENSITIVITY TEST ON CALIBRATION TERM

To further understand the effectiveness of the calibration loss, we perform a sensitivity test of the coefficient $\beta$ in Equation 11. This evaluates how our proposed calibration of parameter mixtures affect the overall model prediction and uncertainty quantification capabilities. We evaluate $\beta$ values of $0, 0.2, 0.5, 0.8$ and $1$, where $\beta = 0$ resembles the original MixLoRA method.

Results in Table 4 demonstrate the effectiveness of the calibration loss. When $\beta = 0$, the model is optimized without calibration on parameter mixtures, resulting in high ECE value. Even with small $\beta = 0.2$ or $\beta = 0.5$, the ECE scores drastically improved compared to no calibration setting. Finally, when $\beta = 1$, the calibration term effectively optimizes the conditional parameter mixtures to generate outputs that fit data distribution well, resulting in lower ECE scores and higher accuracies.

## 6 DISCUSSION & CONCLUSION

In this work, we propose Functional-Level Uncertainty Quantification for Calibrated Fine-Tuning (UQ4CT), which addresses the overconfidence issues commonly encountered during fine-tuning of large language models. We present a functional perspective on quantifying epistemic uncertainty in LLMs and utilize it for uncertainty-calibrated fine-tuning. By incorporating functional-level epistemic uncertainty quantification with a mixture-of-experts framework, our proposed uncertainty-calibrated training loss effectively addresses the challenge of overconfidence in fine-tuned LLMs by significantly improving uncertainty calibration while maintaining high accuracy. Our evaluations demonstrate that UQ4CT reduces the Expected Calibration Error by more than 25% without compromising accuracy across a variety of downstream tasks, including common-sense and domain-specific reasoning, under in-distribution and out-of-distribution scenarios.

The limitation of UQ4CT lies in its dependency on predictive correctness. For general language modeling tasks such as chat completion, there lacks a clear metric on whether the response is correct or not. This limits the application of UQ4CT as naively token matching is a poor indicator of semantic correctness due to the ambiguous nature of language. For future work, we are exploring ways to adapt UQ4CT on open-ended problems that lacks a definitive optimization objective.

REPRODUCIBILITY STATEMENT

We share our experimental details in Appendix A.4, and also provide the code and model weights for running experiments in the supplementary materials to reproduce our model performance results.

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

Table 5: Performance comparison of different methods fine-tuned with Mistral-7B across 4 common sense reasoning tasks and a domain-specific task. UQ4CT shows significant ECE improvements while maintaining high accuracy.

| Metrics | Methods | BoolQ | ARC-E | ARC-C | OBQA | ClimateQA |
|---------|---------|-------|-------|-------|------|-----------|
| ACC ↑ | LoRA | $70.3_{0.62}$ | $84.8_{0.47}$ | $70.2_{0.84}$ | $82.8_{0.62}$ | $72.5_{1.6}$ |
| | MC Drop | $69.6_{1.07}$ | $84.6_{0.91}$ | $69.6_{0.76}$ | $82.6_{0.71}$ | $72.5_{1.6}$ |
| | Ensemble | $71.8_{1.29}$ | $84.2_{0.66}$ | $71.0_{1.41}$ | $82.5_{0.6}$ | $72.9_{2.88}$ |
| | LA | $70.7_{1.82}$ | $82.4_{2.05}$ | $68.5_{3.31}$ | $82.5_{0.77}$ | $71.6_{1.56}$ |
| | MixLoRA | $73.1_{0.38}$ | $85.5_{1.27}$ | $71.2_{1.75}$ | $83.3_{1.14}$ | $72.0_{1.69}$ |
| | UQ4CT | $73.6_{0.28}$ | $85.9_{0.82}$ | $74.4_{0.82}$ | $83.7_{1.22}$ | $73.2_{1.29}$ |
| ECE ↓ | LoRA | $10.17_{0.24}$ | $9.46_{1.62}$ | $18.42_{1.91}$ | $13.3_{0.25}$ | $13.72_{2.62}$ |
| | MC Drop | $10.62_{0.51}$ | $8.91_{1.35}$ | $18.38_{1.66}$ | $13.3_{0.31}$ | $13.72_{2.61}$ |
| | Ensemble | $8.72_{1.13}$ | $8.72_{1.49}$ | $17.0_{0.97}$ | $9.14_{2.82}$ | $12.86_{1.78}$ |
| | LA | $5.33_{2.16}$ | $20.3_{5.7}$ | $21.27_{4.15}$ | $\mathbf{6.41_{3.22}}$ | $14.64_{2.21}$ |
| | MixLoRA | $8.81_{1.03}$ | $8.16_{0.99}$ | $15.51_{3.86}$ | $10.53_{1.73}$ | $14.05_{3.09}$ |
| | UQ4CT | $\mathbf{3.07_{0.83}}$ | $\mathbf{5.7_{0.69}}$ | $\mathbf{7.04_{0.58}}$ | $\underline{7.92_{1.14}}$ | $\mathbf{11.4_{1.14}}$ |

# A APPENDIX

## A.1 LOAD BALANCING LOSS

We follow the load balancing loss in (Li et al., 2024). Given $N$ experts indexed by $i = 1$ to $N$ and a batch $B$ with $T$ tokens, the auxiliary loss is computed as:

$$\mathcal{L}_{aux} = a \cdot N \cdot \sum_{i=1}^{N} \mathcal{F}_i \cdot \mathcal{P}_i, \tag{13}$$

where

$$\mathcal{F}_i = \frac{1}{T} \sum_{x \in B} \mathbb{1}\{argmax_k \mathcal{R}(x)_k = i\}, \mathcal{P}_i = \frac{1}{T} \sum_{x \in B} \mathcal{R}(x)_i. \tag{14}$$

Here, $\mathcal{R}(\cdot)$ is the top-k router, $\mathcal{F}_i$ is the fraction of tokens dispatched to expert $i$ and $\mathcal{P}_i$ is the fraction of the router probability allocated for expert $i$. The final loss is multiplied by the expert count $N$ to keep the loss constant as the number of experts varies, and the constant term $a$ is set to $10^{-2}$ as a multiplicative coefficient, which is large enough to ensure load balancing while remaining small enough not to overwhelm the primary objective.

## A.2 EXPERIMENTAL RESULTS WITH MISTRAL-7B

In this section, we present the results using Mistral-7B (Jiang et al., 2023), a different decoder-based LLM backbone. Table 5 shows the results of fine-tuning Mistral-7B on 4 common-sense reasoning tasks and one domain-specific climate question-answering task.

For each of the tasks, UQ4CT effectively calibrates the parameter mixtures, leading to the best ECE performance in 4 out of 5 tasks. This indicates the robustness of UQ4CT across different LLMs.

## A.3 DECIDING NUMBER OF ACTIVE EXPERTS

One important aspect of the LoRA MoE architecture is how many experts to activate. Here, we investigate the performance impact of different number of active LoRA experts. We evaluate the model performance with 1 to 5 active experts with 8 in total.

As shown in Table 6, 2 active experts give the optimal performance in terms of accuracy and ECE scores. One expert alone cannot capture complicated functional relationships, while more than 2 experts could potentially introduce redundant functional bases to the model, which deviates the output distribution more from data distribution, thus worsening predictive and calibration performance. Additionally, more active experts lead to a more flattened distribution across experts, which hardens the alignment of parameter mixtures during fine-tuning.

Table 6: Performance comparison of UQ4CT with varying number of experts on OBQA dataset. Top-2 expert selection strategy grants best accuracy and calibration.

| Top-K | ACC ↑ | ECE ↓ |
|-------|-------|-------|
| Top-1 | $74.8_{0.62}$ | $7.69_{1.96}$ |
| Top-2 | $\mathbf{77.3_{1.36}}$ | $\mathbf{5.0_{1.15}}$ |
| Top-3 | $75.2_{0.8}$ | $5.8_{0.81}$ |
| Top-4 | $75.8_{0.53}$ | $7.67_{0.46}$ |
| Top-5 | $75.3_{0.5}$ | $6.29_{0.61}$ |

## A.4 TRAINING DETAILS

We train our model with total of $8$ LoRA experts, and select $2$ experts with the highest probability. For each expert, we use $rank = 16$ and $alpha = 32$. We use batch size of 16 to train our model. For climate task, we set the learning rate to $5e - 4$ and dropout rate to $0.1$ to incorporate the small dataset size. For other tasks, we use $2e - 4$ as our learning rate with dropout $0.05$. We use AdamW as our optimizer and a cutoff length of $512$ for prompts during training.

The experimental setup for single LoRA based models is similar with LoRA ranks set to $80$ to accommodate the MoE model size. For the ensemble baseline, we use an ensemble size of $8$ with $rank = 16$. For Laplace-LoRA, we follow the Laplace hyperparameters in this Github Repository.

## A.5 EXPECTED CALIBRATION ERROR

Expected calibration error (ECE) is a commonly used metric to asses uncertainty quantification performance. ECE measures the alignment between prediction accuracy and model confidence through regrouping the predicted probabilities into $m$ bins. This method then computes the weighted average of the difference between average accuracy and confidence in each bin:

$$\text{ECE} = \sum_{m=1}^{M} \frac{|B_m|}{N} |\text{acc}(B_m) - \text{conf}(B_m)|, \tag{15}$$

where $|B_m|$ is the number of evaluated datapoints in bin $m$, acc and conf is calculated as following:

$$\text{acc}(B_m) = \frac{1}{|B_m|} \sum_{i \in B_m} \mathbf{1}(\hat{y}_i = y_i), \tag{16}$$

$$\text{conf}(B_m) = \frac{1}{|B_m|} \sum_{i \in B_m} P(\hat{y}_i). \tag{17}$$

## A.6 MMLU DISTRIBUTION SHIFT DATASET COMPOSITION

- **Computer Science (CS)**:
    - College Computer Science
    - Computer Security
    - High School Computer Science
    - Machine Learning
- **Engineering (Eng)**:
    - Electrical Engineering
- **Law**:
    - International Law
    - Jurisprudence
    - Professional Law
- **Health**:

- Anatomy
- Clinical Knowledge
- College Medicine
- Human Aging
- Nutrition
- Professional Medicine
- Virology

