# OpenReview forum: "Functional-level Uncertainty Quantification for Calibrated Fine-tuning on LLMs"
_ICLR.cc/2025/Conference — Submitted to ICLR 2025_

### Official Review · Reviewer_cPCd · 2024-11-03

**Soundness:** 3
**Presentation:** 2
**Contribution:** 2
**Rating:** 5
**Confidence:** 4

**Summary:**

This paper presents a method for epistemic uncertainty quantification in the context of LoRA fine-tuning, which is also compatible with a mixture-of-experts setup. The authors introduce a specialized training loss, designed to capture the difference between predictive accuracy and functional-level epistemic uncertainty. When applied to the Llama2 model, the proposed method shows improvements across a variety of datasets.

**Strengths:**

The pipeline and method developed in this paper appear well-founded, and the paper is generally well-written.

**Weaknesses:**

The biggest weakness of this paper is the limited experimental effort. Specifically, the experiments are conducted on only one LLM, and there is very little ablation or other study to investigate the additional properties of this UQ method. While I don’t consider the number of pages a deterministic measure of effort, the brevity of the experimental section does suggest a lack of thoroughness in the design of experiments.

Additionally, beyond the lack of comprehensiveness, the paper offers very little empirical or theoretical insight into why this approach is effective. I also find that the paper lacks depth overall.

**Questions:**

1. In equation (9), what is the term $\mathcal{L}_b$? I don't think this is a good practice to put it in the reference. It should be at least written in the appendix.

2. In table 1 and 2, what is the numeric values in the subscript?

---

> ### Author Response · Authors · 2024-11-23
> **Response to Reviewer cPCd (1/2)**
>
> We thank the reviewer for the insightful questions and helpful suggestions for experimental comprehensiveness. We address the concerns as follows:
>
> >The biggest weakness of this paper is the limited experimental effort. Specifically, the experiments are conducted on only one LLM, and there is very little ablation or other study to investigate the additional properties of this UQ method. While I don’t consider the number of pages a deterministic measure of effort, the brevity of the experimental section does suggest a lack of thoroughness in the design of experiments.
>
> We appreciate the suggestions on empirical evaluations and have added multiple experiments and ablation studies to assess our method (UQ4CT) more comprehensively. For all experiments, we evaluate performance with 3 random runs and report the mean and standard deviations.
>
> __Baseline Comparisons with Mistral-7B.__ The experimental results demonstrates that our method is applicable to different LLMs, outperforming existing methods in uncertainty calibration while maintaining accuracy:
>
> | Metrics | Methods | BoolQ | ARC-E | ARC-C | OBQA | ClimateQA |
> |---|---|---|---|---|---|---|
> | ACC $\uparrow$ | LoRA | $70.3_{0.62}$ | $84.8_{0.47}$ | $70.2_{0.84}$ | $82.8_{0.62}$ | $72.5_{1.6}$ |
> |  | MC Drop | $69.6_{1.07}$ | $84.6_{0.91}$ | $69.6_{0.76}$ | $82.6_{0.71}$ | $72.5_{1.6}$ |
> |  | Ensemble | $71.8_{1.29}$ | $84.2_{0.66}$ | $71.0_{1.41}$ | $82.5_{0.6}$ | $72.9_{2.88}$ |
> |  | LA | $70.7_{1.82}$ | $82.4_{2.05}$ | $68.5_{3.31}$ | $82.5_{0.77}$ | $71.6_{1.56}$ |
> |  | MixLoRA | $73.1_{0.38}$ | $85.5_{1.27}$ | $71.2_{1.75}$ | $83.3_{1.14}$ | $72.0_{1.69}$ |
> | | UQ4CT | $73.6_{0.28}$ | $85.9_{0.82}$ | $74.4_{0.82}$ | $83.7_{1.22}$ | $73.2_{1.29}$ |
> | ECE $\downarrow$ | LoRA | $10.17_{0.24}$ | $9.46_{1.62}$ | $18.42_{1.91}$ | $13.3_{0.25}$ | $13.72_{2.62}$ |
> |  | MC Drop | $10.62_{0.51}$ | $8.91_{1.35}$ | $18.38_{1.66}$ | $13.3_{0.31}$ | $13.72_{2.61}$ |
> |  | Ensemble | $8.72_{1.13}$ | $8.72_{1.49}$ | $17.0_{0.97}$ | $9.14_{2.82}$ | $12.86_{1.78}$ |
> |  | LA | $5.33_{2.16}$ | $20.3_{5.7}$ | $21.27_{4.15}$ | $\mathbf{6.41_{3.22}}$ | $14.64_{2.21}$ |
> |  | MixLoRA | $8.81_{1.03}$ | $8.16_{0.99}$ | $15.51_{3.86}$ | $10.53_{1.73}$ | $14.05_{3.09}$ |
> | | UQ4CT | $\mathbf{3.07_{0.83}}$ | $\mathbf{5.7_{0.69}}$ | $\mathbf{7.04_{0.58}}$ | $\underline{7.92_{1.14}}$ | $\mathbf{11.4_{1.14}}$ |
>
>
> __Ablation Study: # of Experts to Select.__ We conduct an ablation study on the number of experts to select on OBQA dataset with Llama2-7B. This explores how the number of active experts can impact the functional relationships utilized by the model for inference:
>
> | Top-K | ACC $\uparrow$ | ECE $\downarrow$ |
> |---|---|---|
> | Top-1 | $74.8_{0.62}$ | $7.69_{1.96}$ |
> | Top-2 | $\mathbf{77.3_{1.36}}$ | $\mathbf{5.0_{1.15}}$ |
> | Top-3 | $75.2_{0.8}$ | $5.8_{0.81}$ |
> | Top-4 | $75.8_{0.53}$ | $7.67_{0.46}$ |
> | Top-5 | $75.3_{0.5}$ | $6.29_{0.61}$ |
>
>  An theoretical insight of using the top 2 experts out of 8 in total is that one expert alone cannot capture complicated functional relationships, while more than 2 experts introduce redundant functional bases to the model, which deviates the output distribution more from data distribution, thus worsening predictive and calibration performance.
>
>
> __Ablation Study: Varying Loss Coefficients.__ We also conduct an ablation study on varying loss terms where we fix alpha (coefficient of the load balancing term) to 1 and compare model performance with different beta values (0, 0.2, 0.5, 0.8, 1) on OBQA dataset with Llama2-7B:
>
> | $\beta$ | ACC $\uparrow$ | ECE $\downarrow$ |
> |---|---|---|
> | 0 | $75.5_{2.91}$ | $12.9_{1.99}$ |
> | 0.2 | $76.0_{0.6}$ | $7.47_{0.78}$ |
> | 0.5 | $75.9_{0.31}$ | $7.82_{0.93}$ |
> | 0.8 | $76.6_{0.4}$ | $5.94_{1.16}$ |
> | 1 | $\mathbf{77.3_{1.36}}$ | $\mathbf{5.0_{1.15}}$ |
>
> The results demonstrate the effectiveness of our proposed calibration loss. When $\beta = 0$, the model is optimized without calibration on parameter mixtures, resulting in high ECE value. Even with small $\beta=0.2$ or $\beta=0.5$, the ECE scores drastically improved compared to no calibration setting. Finally, when $\beta=1$, the calibration term effectively optimizes the conditional parameter mixtures to generate outputs that fit data distribution well, resulting in lower ECE scores and higher accuracies.

---

> ### Author Response · Authors · 2024-11-23
> **Response to Reviewer cPCd (2/2)**
>
> >Additionally, beyond the lack of comprehensiveness, the paper offers very little empirical or theoretical insight into why this approach is effective. I also find that the paper lacks depth overall.
>
> For empirical insights, we evaluate UQ4CT and the baselines with a diverse set of tasks, covering both common-sense reasoning (i.e., OBQA, BOOLQ) and domain-specific tasks(i.e., ClimateQA). The strong empirical results across tasks showing over 25% reductions in Expected Calibration Error (ECE) compared to baselines as shown in Tables 1 and 2 underscores the generalizability of UQ4CT and validates its applicability across different domains and contexts.
>
> These results collectively provide robust empirical insights into the effectiveness and practicality of UQ4CT. Additionally, we have added experimental results with Mistral-7B and multiple ablation studies to enhance comprehensiveness.
>
> We have demonstrated theoretical insights in Section 3.1, 3.3 and also Figure 1 in our paper. The calibration loss effectively promotes expert exploitation for correct predictions and expert exploration for incorrect predictions. Ideally, when the $K$ LoRA experts together have captured all the functional relationships across the data distribution after fine-tuning (via cross entropy loss), our proposed loss $\mathcal{L}_\{cal}$ calibrates the mixture of LoRA experts conditioned on the input $x$ across the training set. This ensures that the model effectively selects the appropriate experts to produce outputs aligned with the underlying data distribution corresponding to $x$.
>
> Incorporating the revision suggestions and additional experiments contributes to the depth of our work, as the new evaluations with Mistral-7B and comprehensive ablation studies ensure a robust foundation for our claims.
>
> >In equation (9), what is the term $\mathcal{L}_b$? I don't think this is a good practice to put it in the reference. It should be at least written in the appendix.
>
> We appreciate the suggestion. We have included the definition of $\mathcal{L}_b$ in the appendix in the revised version. The term $\mathcal{L}\_b$ is the load balancing term proposed in [1] to address the phenomenon that some experts tend to be chosen more frequently by the top-k routers during fine-tuning.
>
> Given $N$ experts indexed by $i=1$ to $N$ and a batch $B$ with $T$ tokens, the auxiliary loss is computed as:
> $\mathcal{L}\_{aux}=a \cdot N \cdot \sum^{N}\_{i=1}\mathcal{F}\_i \cdot \mathcal{P}\_i$,
>
> where:
>
> $\mathcal{F}\_i = \frac{1}{T}\sum_{x\in B} \mathbb{1}\\{argmax_k \mathcal{R}(x)_k=i\\}, \mathcal{P}_i = \frac{1}{T}\sum\_{x\in B}\mathcal{R}(x)_i$.
>
> Here, $\mathcal{R}(\cdot)$ is the top-k router, $\mathcal{F}_i$ is the fraction of tokens dispatched to expert $i$ and $\mathcal{P}_i$
> is the fraction of the router probability allocated for expert $i$. The final loss is multiplied by the expert count $N$ to keep the loss constant as the number of experts varies, and the constant term $a$ is set to $10^{-2}$as a multiplicative coefficient, which is large enough to ensure load balancing while remaining small enough not to overwhelm the primary objective.
>
> >In table 1 and 2, what is the numeric values in the subscript?
>
> We apologize for the confusion. We report the mean and standard deviations across 3 random runs for each experiment. The subscript corresponds to the standard deviations. We have added explanations in the revised version.
>
> [1] Li, Dengchun, et al. "Mixlora: Enhancing large language models fine-tuning with lora based mixture of experts."

---

> ### Comment · Reviewer_cPCd · 2024-11-27
> **Response**
>
> Thanks for the response. My concerns are partially addressed and I will raise the score from 3 to 5.

---

### Official Review · Reviewer_d11o · 2024-11-04

**Soundness:** 3
**Presentation:** 3
**Contribution:** 3
**Rating:** 8
**Confidence:** 3

**Summary:**

The authors propose a novel calibration method for LLMs that is applicable already during the fine-tuning stage of the model. This is achieved by leveraging the specific nature of the LoRA MoE architecture, using the weights of the Top-K router layers to quantify the functional epistemic uncertainty. The epistemic uncertainty calculated this way is used with the predictive accuracy in a specific loss function (as one part of a triple loss function for the overall loss). The authors claim to reduce ECE substantially while not impairing the predictive performance, but even resulting in better predictive performance and making it more robust to e.g. distribution shift. These findings are backed by two experiments: First, they show ECE decrease across 5 MCQA benchmarks while maintaining better (or at least on par) accuracy compared to the baselines. Second, they use a subset of the benchmarks to create distribution shift scenarios to prove the robustness of their approach.

**Strengths:**

The experiments are outlined very clearly and explained and motivated comprehensively. This makes the paper easy to follow and understandable to the reader. The additional experiments (distribution shift and ablation) are well-motivated and contribute to the overall experimental setup.
- The problem tackled in this research is widely known and highly problematic, yet there is no definite and generally applicable solution to it. Hence this work, tackling it effectively for LoRA MoE architectures is an important contribution to the field in general.
- Not only do the authors introduce a method that can substantially improve the model calibration, but it also shows strong model performance in terms of accuracy.

**Weaknesses:**

- The ablation study/studies could be more elaborated, i.e. e.g. examining what the influence of the different terms in the loss is, or how sensitive performance is to the hyperparameters alpha and beta in the loss. I think such examinations are important to understand what’s going on and how brittle such triple losses are.
- The approach is specifically tailored to the nature of the LoRA MoE architecture as it critically depends on leveraging the weights of the Top-K Routing layers of the MoE. While this seems to work pretty well here, one might also consider this a limitation of the approach in general since it is not generally applicable.

**Questions:**

- In Equation (5): What exactly do M’ and a’ refer to? I believe this is not explicitly specified anywhere
- Why did you place the “Related work” section after your methodology? I think it introduces important concepts to the reader who would thus benefit from reading this before your explanations in the “Methodology” section.
- L. 326: I personally think it’s not optimal to call something “significant” that you actually did not formally test with a statistical test. Maybe it’s better to use a formulation like “notable” or “substantial”?
- L 359: Just to be sure, what do you mean by “bin size of 15”? I think it is more common to communicate this in terms of “number of bins (of equal size usually”?
- L. 447 More of a comment than a question, but I think a sole subsubsection (5.4.1) does not really make sense, probably you should rather just add this subsubheading to the heading of subsection 5.4.

---

> ### Author Response · Authors · 2024-11-23
> **Response to Reviewer d11o**
>
> We appreciate the constructive feedback and thoughtful suggestions. We address the concerns as follows:
>
> >The ablation study/studies could be more elaborated, i.e. e.g. examining what the influence of the different terms in the loss is, or how sensitive performance is to the hyperparameters alpha and beta in the loss. I think such examinations are important to understand what’s going on and how brittle such triple losses are.
>
> This is a great suggestion, we have added an ablation study on varying loss terms where we fix alpha (coefficient of the load balancing term) to 1 and compare model performance with different beta values (0, 0.2, 0.5, 0.8, 1) on OBQA dataset with Llama2-7B. Results are given as following, we report the mean and standard deviation of the performance under 3 random runs:
> | $\beta$ | ACC $\uparrow$ | ECE $\downarrow$ |
> |---|---|---|
> | 0 | $75.5_{2.91}$ | $12.9_{1.99}$ |
> | 0.2 | $76.0_{0.6}$ | $7.47_{0.78}$ |
> | 0.5 | $75.9_{0.31}$ | $7.82_{0.93}$ |
> | 0.8 | $76.6_{0.4}$ | $5.94_{1.16}$ |
> | 1 | $\mathbf{77.3_{1.36}}$ | $\mathbf{5.0_{1.15}}$ |
>
>
> The results demonstrate the effectiveness of our proposed calibration loss. When $\beta = 0$, the model is optimized without calibration on parameter mixtures, resulting in high ECE value. Even with small $\beta=0.2$ or $\beta=0.5$, the ECE scores drastically improved compared to no calibration setting. Finally, when $\beta=1$, the calibration term effectively optimizes the conditional parameter mixtures to generate outputs that fit data distribution well, resulting in lower ECE scores and higher accuracies. We will add this ablation study in the revised version.
>
> >The approach is specifically tailored to the nature of the LoRA MoE architecture as it critically depends on leveraging the weights of the Top-K Routing layers of the MoE. While this seems to work pretty well here, one might also consider this a limitation of the approach in general since it is not generally applicable.
>
> Although our approach depends on the LoRA MoE architecture for LLMs, the core ideas of functional-level epistemic uncertainty quantification and calibration can indeed be generalized and applied to broader settings beyond LoRA. For example, [1] explores the MoE architecture with various deep learning networks such as CNN/ResNet, demonstrating the broad applicability of MoE architecture. The MoE mechanism leverages router distribution to dynamically assign inputs to different experts based on the hidden states. This is not exclusive to LoRA and can be applied to other neural networks, including non-transformer architectures.
>
> >In Equation (5): What exactly do M’ and a’ refer to? I believe this is not explicitly specified anywhere
>
> We appreciate the feedback. We have added demonstrations in the paper accordingly. M’ refers to the optimal mixture of experts with model M, and a’ refers to the ground truth answer generated by model M’.
>
> >Why did you place the “Related work” section after your methodology? I think it introduces important concepts to the reader who would thus benefit from reading this before your explanations in the “Methodology” section.
>
> We presented the 'Preliminaries' section prior to the 'Method' section, demonstrating the necessary and context-specific background knowledge required to understand our proposed method. After 'Method' section, we then present the 'Related Work' section for a broader overview of recent advancements in relevant fields, offering insights that inform the selection of experimental baselines.
>
> >L. 326: I personally think it’s not optimal to call something “significant” that you actually did not formally test with a statistical test. Maybe it’s better to use a formulation like “notable” or “substantial”?
>
> >L 359: Just to be sure, what do you mean by “bin size of 15”? I think it is more common to communicate this in terms of “number of bins (of equal size usually”?
>
> We apologize for the ambiguity. We have revised the wording in the paper. Here it should be “number of bins” instead of “bin size”.
>
> >L. 447 More of a comment than a question, but I think a sole subsubsection (5.4.1) does not really make sense, probably you should rather just add this subsubheading to the heading of subsection 5.4.
>
> Thanks for the suggestion, it is reasonable to do so with only one ablation study. Since we have added multiple ablation studies now, we are allocating them into multiple subsections.
>
>
>
> [1] Chen, Zixiang, et al. "Towards Understanding the Mixture-of-Experts Layer in Deep Learning."

---

> > ### Comment · Reviewer_d11o · 2024-11-25
> > **Thanks for you answer**
> >
> > Thank you very much for answering my questions and clarifying the ambiguities.

---

### Official Review · Reviewer_ydTf · 2024-11-04

**Soundness:** 3
**Presentation:** 3
**Contribution:** 2
**Rating:** 5
**Confidence:** 3

**Summary:**

This paper proposes a new approach called Functional-Level Uncertainty Quantification for Calibrated Fine-Tuning (UQ4CT) to calibrate the functional-level epistemic uncertainty via the LoRA MoE architecture.

**Strengths:**

1. This paper introduces a unique uncertainty quantification (UQ) method for fine-tuning large language models (LLMs) using a Mixture-of-Experts (MoE) approach
2. In this paper, UQ4CT reportedly achieves a 25% reduction in Expected Calibration Error (ECE) across multiple benchmarks, which is a substantial improvement.

**Weaknesses:**

1. The novel calibration loss function is mentioned as one of the contributions. However, the paper lacks detailed theoretical analysis to illustrate why this specific design especially the loss is effective for uncertainty calibration.
2. Although the paper compares UQ4CT with other PEFT-based uncertainty quantification techniques, a more comprehensive comparison with non-PEFT-based methods might provide a clearer view of its advantages and limitations.
3. The whole framework is built on LoRA, and can it be applied to more general settings?
4. There are a lot of hyper-parameters, such as the number of top gate routers (why use 2 in this paper), and N, s, L. How to set them in practice and is there any theoretical insight for these choices?

**Questions:**

Please refer to the weakness part.
For example:
1. Can the authors add some theoretical analysis?
2. Can more baseline methods be compared?

---

> ### Author Response · Authors · 2024-11-23
> **Response to Reviewer ydTf (1/2)**
>
> We appreciate the constructive feedback and thoughtful suggestions and address the concerns as follows:
>
> >The novel calibration loss function is mentioned as one of the contributions. However, the paper lacks detailed theoretical analysis to illustrate why this specific design especially the loss is effective for uncertainty calibration.
>
> >Can the authors add some theoretical analysis?
>
> We demonstrate the theoretical insight of our proposed method as following:
>
> When the $K$ LoRA experts together have captured all the functional relationships across the data distribution after fine-tuning (via cross entropy loss), and the training and testing data share the same distribution, our proposed loss $\mathcal{L}_\{cal}$ calibrates the mixture of LoRA experts conditioned on the input $x$ across the training set. This ensures that the model effectively selects the appropriate experts to produce outputs aligned with the underlying data distribution corresponding to $x$.
>
> We will add detailed proof in the revised version. Empirically, our results show that UQ4CT effectively aligns the model's predictive confidence with the underlying data distribution by selecting the appropriate functional bases conditioned on the input $x$ after calibration.
>
> >Although the paper compares UQ4CT with other PEFT-based uncertainty quantification techniques, a more comprehensive comparison with non-PEFT-based methods might provide a clearer view of its advantages and limitations.
>
> Thanks for the suggestion, we have explored recent non-PEFT-based uncertainty quantification methods on LLMs while none of them applies to our setup. Current non-PEFT-based uncertainty quantification methods focus on open-ended questions while current PEFT-based methods are limited to multiple-choice questions. Out of three recent works on non-PEFT-based methods [1,2,3], only [1] is applicable to multiple-choice questions with the question-paraphrasing approach. [2] requires multiple high-quality clarification phrases for each prompt, while [3] relies on multiple samples given the same question, which does not fit into the multiple-choice scenario.
>
> Although [1] is applicable, they conducted the experiment with models that are orders of magnitude bigger than our setup (ChatGPT3.5/ChatGPT4, etc). When extending their method to smaller models without fine-tuning, their approach of perturbing the questions and the temperature significantly worsens the predictive accuracy, thus making the calibration measurements meaningless. We provide the results with [1] in the following table just for demonstration purposes. The results are obtained with Llama2-7b-hf:
>
> | Metrics | BoolQ | ARC-E | ARC-C | OBQA | ClimateQA |
> |---|---|---|---|---|---|
> | ACC $\uparrow$  | 19.11 | 23.44 | 27.72 | 28.73 | 17.39 |
> | ECE $\downarrow$  | 64.46 | 37.58 | 33.45 | 30.6 | 52.54 |
>
> >The whole framework is built on LoRA, and can it be applied to more general settings?
>
> Absolutely, while the framework described in the paper is built on the LoRA Mixture-of-experts (MoE) architecture, the core ideas of functional-level epistemic uncertainty quantification and calibration can indeed be generalized and applied to broader settings beyond LoRA. For example, [4] explores the MoE architecture with various deep learning networks such as CNN/ResNet, demonstrating the broad applicability of MoE architecture. The MoE mechanism leverages router distribution to dynamically assign inputs to different experts based on the hidden states. This is not exclusive to LoRA and can be applied to other neural networks, including non-transformer architectures.
>
> [1] Gao, Xiang, et al. "SPUQ: Perturbation-Based Uncertainty Quantification for Large Language Models."
>
> [2] Hou, Bairu, et al. "Decomposing Uncertainty for Large Language Models through Input Clarification Ensembling."
>
> [3] Kuhn, Lorenz, et al. "Semantic Uncertainty: Linguistic Invariances for Uncertainty Estimation in Natural Language Generation."
>
> [4] Chen, Zixiang, et al. "Towards Understanding the Mixture-of-Experts Layer in Deep Learning."

---

> ### Author Response · Authors · 2024-11-23
> **Response to Reviewer ydTf (2/2)**
>
> >There are a lot of hyper-parameters, such as the number of top gate routers (why use 2 in this paper), and N, s, L. How to set them in practice and is there any theoretical insight for these choices?
>
> Generally, we follow the hyperparameter setup in previous works regarding cutoff length and total number of LoRA experts. For the optimizer hyperparameters, we perform a hyperparameter search and use the optimal value found. For the number of top gate routers, we performed an exploration from 1 to 5 experts and found that 2 is the optimal. The performance comparison on the OBQA dataset with Llama2-7B is given as following, we report the mean and standard deviation of the performance under 3 random runs:
>
> | Top-K | ACC $\uparrow$ | ECE $\downarrow$ |
> |---|---|---|
> | Top-1 | $74.8_{0.62}$ | $7.69_{1.96}$ |
> | Top-2 | $\mathbf{77.3_{1.36}}$ | $\mathbf{5.0_{1.15}}$ |
> | Top-3 | $75.2_{0.8}$ | $5.8_{0.81}$ |
> | Top-4 | $75.8_{0.53}$ | $7.67_{0.46}$ |
> | Top-5 | $75.3_{0.5}$ | $6.29_{0.61}$ |
>
> We will also add this as an ablation study in the revised version. An theoretical insight of using the top 2 experts out of 8 in total is that one expert alone cannot capture complicated functional relationships, while more than 2 experts introduce redundant functional bases to the model, which deviates the output distribution more from data distribution, thus worsening predictive and calibration performance.

---

> ### Comment · Reviewer_ydTf · 2024-11-26
>
> Thanks for answering my questions but I still think that some theoretical analysis is necessary

---

### Official Review · Reviewer_3vnu · 2024-11-06

**Soundness:** 3
**Presentation:** 3
**Contribution:** 3
**Rating:** 6
**Confidence:** 3

**Summary:**

This paper introduces a novel method called Functional-Level Uncertainty Quantification for Calibrated Fine-Tuning (UQ4CT), aimed at addressing overconfidence in large language models (LLMs) during fine-tuning, especially in scenarios with limited or sparse data. Traditional parameter-efficient fine-tuning (PEFT) approaches often fail to accurately calibrate epistemic (model-related) uncertainty, leading to overly confident predictions. To overcome this, UQ4CT incorporates a Mixture-of-Experts (MoE) framework with LoRA (Low-Rank Adaptation), enabling dynamic, prompt-dependent expert selection that captures and calibrates functional-level uncertainty throughout the fine-tuning process. The paper’s main contributions are:
1. A new approach to quantify epistemic uncertainty at the functional level during fine-tuning, using MoE to dynamically adjust uncertainty based on input.
2. A calibration loss function that aligns uncertainty with predictive correctness, encouraging expert exploration for incorrect predictions and reinforcing correct predictions.
3. Empirical results showing that UQ4CT reduces Expected Calibration Error (ECE) by over 25% across multiple benchmarks while maintaining high accuracy, demonstrating robustness both within distribution and under distribution shifts.

**Strengths:**

The paper introduces a novel approach by integrating functional-level epistemic uncertainty directly into the fine-tuning process of LLMs, departing from conventional post hoc calibration. The unique combination of MoE with LoRA enables dynamic, prompt-dependent uncertainty modeling, addressing both parameter efficiency and overconfidence in fine-tuned models. The proposed calibration loss function further aligns model confidence with predictive accuracy, reducing overconfidence in a novel, functional manner.
The paper is validated with comprehensive experiments across multiple benchmarks, supported by baseline comparisons and ablation studies.
The paper is well-structured and accessible, with clear explanations and effective use of visual aids, making complex concepts understandable.
UQ4CT addresses a critical need for reliable uncertainty estimation in high-stakes applications by enhancing calibration and robustness under distribution shifts. This advancement strengthens the foundation for trustworthy AI, especially in domains where reliable predictions are essential despite sparse data.

**Weaknesses:**

UQ4CT’s calibration loss relies on clear correctness metrics, limiting its applicability to tasks with definitive right or wrong answers. This restricts its use in open-ended tasks, such as generative dialogue or summarization. To broaden applicability, future work could explore calibration methods based on human feedback scores or soft accuracy measures instead of binary correctness.
The use of MoE for dynamic expert selection, while effective, could lead to scalability issues with larger models or multi-task scenarios due to increased computational demands. Optimizing expert selection with pruning or adaptive sparsity methods could help make UQ4CT more efficient and feasible for resource-limited applications.

**Questions:**

1. Since UQ4CT’s calibration relies on binary correctness, how might this approach be adapted for tasks where “correctness” is less definitive, such as open-ended text generation or summarization?
2. Could the authors elaborate on any experiments or initial explorations in applying UQ4CT to such tasks?
3. Exploring approaches that rely on alternative correctness signals, such as soft scoring or feedback-based calibration, might broaden the applicability.

---

> ### Author Response · Authors · 2024-11-22
> **Response to Reviewer 3vnu**
>
> We appreciate the constructive feedback and thoughtful suggestions. We address the concerns as follows:
> >The use of MoE for dynamic expert selection, while effective, could lead to scalability issues with larger models or multi-task scenarios due to increased computational demands. Optimizing expert selection with pruning or adaptive sparsity methods could help make UQ4CT more efficient and feasible for resource-limited applications.
>
> Optimizing with pruning would definitely help with reducing computational resources needed. However, the LoRA Mixture of Experts (MoE) architecture is highly unlikely to experience scalability issues because of its parameter-efficient nature. As reported in [2], the MoE architecture only introduces 2.9% additional parameters regarding the Llama2-7B model, which has 7 billion parameters. For multi-task scenarios, some existing applications [3,4] with similar LoRA MoE architecture have shown promising results without scalability issues. This is because the number of experts is independent from different problem difficulties and, according to the empirical results in [3,4], does not scale up even for multi-task scenarios.
>
> Computational wise, LoRA MoE only inherently activates a small subset of experts for each input (i.e. 2 out of 8 LoRA experts in our setup), which reduces the computational overhead compared to dense approaches. This sparsity ensures that even with a large number of experts, the computational cost scales more efficiently than with a dense approach of equivalent size.
>
>
> >Since UQ4CT’s calibration relies on binary correctness, how might this approach be adapted for tasks where “correctness” is less definitive, such as open-ended text generation or summarization?
>
> For open-ended text generation / summarization problems, there indeed lacks a definitive optimization objective. We can adapt UQ4CT to open-ended text generation or summarization tasks with non-definitive signals, such as human preference feedback for RLHF tasks or semantic uncertainty [1], which defines and utilizes semantic entropy in detecting hallucinations in open-ended question answering tasks.
>
> >Could the authors elaborate on any experiments or initial explorations in applying UQ4CT to such tasks?
>
> >Exploring approaches that rely on alternative correctness signals, such as soft scoring or feedback-based calibration, might broaden the applicability.
>
> This is a great point. We are actively exploring methods to apply UQ4CT for open-ended question-answering datasets. The semantic entropy method in [1] is a suitable alternative to the predictive correctness for open-ended tasks. Our current approach is to compute the semantic entropy following [1] and use it as the target score to calibrate the functional level uncertainty. Considering the rebuttal time limitation and implementation complexity, we would not be able to show the performance of this approach.
>
> [1] Kuhn, Lorenz, et al. "Semantic Uncertainty: Linguistic Invariances for Uncertainty Estimation in Natural Language Generation."
>
> [2] Li, Dengchun, et al. "Mixlora: Enhancing large language models fine-tuning with lora based mixture of experts."
>
> [3] Liu, Qidong, et al. “When MOE Meets LLMs: Parameter Efficient Fine-tuning for Multi-task Medical Applications.”
>
> [4] Feng, Wenfeng, et al. “Mixture-of-LoRAs: An Efficient Multitask Tuning for Large Language Models.”

---

### Author Response · Authors · 2024-11-26
**Summary of Rebuttal Revision**

We thank all the reviewers for the insightful questions and the positive, high-quality feedback, which has helped us to improve the paper significantly.

We appreciate all reviewers for pointing out that our method is novel and well-motivated and the writing is clear. We have carefully addressed each question, revised our paper with the feedbacks and added the valuable experiments requested for more comprehensive assessment of our method.

To summarize, we have uploaded a revised version with the following main additions (marked in blue on the paper):

__Additional Experiments__:
1. In-distribution performance evaluation with Mistral-7B on $5$ tasks across $5$ baseline methods and our method (Table 5).
2. Sensitivity test of our proposed calibration loss on $5$ different weighting setups from $0$ to $1$ (Table 4).
3. Performance with different number of active experts from $1$ to $5$ (Table 6).

__More Theoretical Insights__:
1. We highlight the functional perspective of LoRA Mixture-of-Experts and their resemblance of epistemic uncertainty (3.1, 3.2).
2. We further explain how our approach calibrates the mixture of LoRA Experts given input $x$ by conditionally promotes expert exploration and exploitation (3.3).

We address the individual questions of each reviewer below in specific replies.


---
Please do not hesitate to inform us if you have any additional concerns or suggestions. We sincerely appreciate your valuable insights and feedbacks throughout the review process. Thank you once again for your time and effort in reviewing our work.

---

### Meta-Review · Area_Chair_y9wY · 2024-12-22

**Metareview:**

This paper aims to improve the issue of overconfidence in fine-tuned LLMs, and proposes the functional-level uncertainty quantification for calibrated fine-tuning (UQ4CT), which captures and calibrates functional-level epistemic uncertainty during the fine-tuning stage via a mixture-of-expert framework. The reviewers have a mixed opinion, and the main concerns (Reviewers ydTf and cPCd) come from the fact that the empirical and theoretical insights on why the method works are not presented with good depth. I somewhat agree with these two reviewers, and therefore recommend the authors revise and resubmit to the next conference.

**Additional Comments On Reviewer Discussion:**

Reviewers all engaged during the rebuttal phases, and some concerns remain.

---

### Decision · Program_Chairs · 2025-01-22

Reject